

# Field Calibration of Low-Cost Air Pollution Sensors

Andres Gonzalez[1], Adam Boies[2], Jacob Swason[3], David Kittelson[4]

[1] Department of Civil, Environmental, and Geo- Engineering, University of Minnesota, Minneapolis, MN 55455, USA
[2] Department of Engineering, University of Cambridge, Cambridge, CB2 1PZ, UK
5 [3] Department of Integrated Engineering, Minnesota State University, Mankato, MN 56001, USA
[4] Department of Mechanical Engineering, University of Minnesota, Minneapolis, MN 55455, USA

*Correspondence to*: Andres Gonzalez (gonza918@umn.edu)

**Abstract.**

10   To implement effective policies and strategies to control air pollution, it is crucial to obtain accurate air quality data. Stationary air monitoring stations (AMSs) help local authorities and environmental agencies in achieving these goals; however, these measurements have limitations. AMSs provide detailed temporal data on air quality, but only at discrete locations at relatively high cost. An alternative method, low-cost mobile air quality monitoring (LCMAQM) sensors, complement AMSs. LCMAQM sensors can cover larger areas and the cost of typical sensors for LCMAQM are $150-200 each. We have developed a wireless Mobile Autonomous Air Quality Sensor box (MAAQSbox) to measure air pollution. The MAAQSbox contains LCMAQM sensors (gas and particle) and a wireless broadcasting system, which enables autonomous field operation for varied mobile applications. Nitrogen dioxide ($NO_2$), nitric oxide (NO), carbon monoxide (CO), and ozone ($O_3$) gases are measured by B4 sensors. Particulate matter ($PM_{2.5}$) is measured by OPC-N2. A field calibration has been performed by making side by side measurements with the MAAQSbox and Minnesota Pollution Agency AMS. The calibrations of LCMAQM sensors were 20   determined by multivariate linear regressions (MLR). MLR results for all sensors were improved by including the temperature and relative humidity as independent variables. The $R^2$ of CO, NO, $NO_2$, and $O_3$ gas sensors are 0.96, 0.97, 0.81, and 0.95 respectively, while the $R^2$ of $PM_{2.5}$ particle sensor is 0.6. B4 sensors are sensitive to ambient conditions such as temperature and relative humidity. The results with OPC-N2 differs from the AMS indicating further developments are needed to enable more accurate $PM_{2.5}$ measurements.

## 1 Introduction

Air pollution in urban areas is one of the main health issues cities face. According to WHO, more than 80% of urban inhabitants are exposed to a high level of air pollution increasing the risk of stroke, heart disease, lung cancer, and chronic and acute respiratory diseases. Also, increases in morbidity and mortality due to extreme air pollution episodes are very well-documented (Dockery et al., 1993). Today, the number of estimated deaths, as a result of air outdoor pollution, around the world is 30   approximately 3.3 million per year (Lim et al., 2010); this number is estimated to double by 2050 (Lelieveld et al., 2015).



To implement effective policies and strategies to control air pollution, it is crucial to obtain accurate air quality data. These data are essential to assess the public health impact of air pollution and to evaluate the effectiveness of any plan to address the problem. Stationary air monitoring stations (AMS) help us in achieving these goals; however, these measurements have limitations. AMS provide data on air quality in only one specific location, meaning there are uncertainties about air quality

nearby (California Environmental Health, 2017). Also, AMS measurements may not represent human street-level exposures, as the AMS sample inlet heights can be greater than 3 m (EPA, 2018). Given the large heterogeneity of air pollution within a roadway and nearby, the significant differences in pollution throughout an urban environment are not captured by single-location measurements. The second limitation is the cost of an AMS, which can reach $150,000-200,000 or more (UNEP, 2015). This is an important limitation for local governments where budgetary constraints limit the number of AMS sites.

An alternative method, low-cost, mobile air quality monitoring (LCMAQM) sensors, can complement AMS measurements. First, LCMAQM sensors can cover larger stationary areas, and thus supplement AMS data. They can also obtain more spatial resolution in air quality measurements since mobile sensors can be placed in a moving car, truck, train, bus, or other mobile sources. Second, the cost of typical sensors for LCMAQM are only $150-200 each; thus, this technology is more accessible for local governments or authorities, crucially in developing countries where pollution levels are typically the highest (Arceo

et al., 2016).

However, the LCMAQM technology has limitations. The data obtained by these sensors are less reliable compared to the data collected by sensors from AMS (Schneider et al., 2017). Environmental conditions, mainly ambient temperature and humidity (Rai et al., 2017) affect the output of LCMAQM sensors. Thus, there are challenges to using this sensor technology in the field where it is impossible to control temperature and relative humidity. In periods of high temperatures, LCMAQM sensors diverge

from the reference sensors affecting the results of measurements (Castell et al., 2017; Cross et al., 2018; Hagan et al., 2018; Afshar-Mohajer et al., 2018; Mijiling et al., 2017). Similarly, significant errors in measured concentration have been reported at relative humidity higher than 77% (Casey et al., 2018; Afshar-Mohajer et al., 2018; Mijiling et al., 2017; Wei et al., 2018). Consequently, ambient temperature and humidity levels should be included in the algorithms that relate the sensor output to concentrations.

There are also cross-sensitivities of LCMAQM sensors with other gases, but they vary by sensor type (Castell et al., 2017). The response of each sensor differs; thus, field calibration is required to obtain valid data (Castell et al., 2017).

There are several projects using the low-cost sensors in a mobile platform. One of the first research projects in LCMAQM was OpenSense implemented in Zurich and Lausanne, Switzerland using buses, trams, and electrical cars (OpenSense, 2014). Citi-Sense-MOB is another LCMAQM project developed and implemented in Oslo, Norway using buses and electrical bicycles as

the platforms (Castell et al., 2014). Then this project was extended to seven other cities in Europe (Fishbain et al., 2017). A third project took place in Bari, Italy in which a sensor array was mounted in a bus to collect data achieving promising results in air quality measurements (Penza et al., 2017). The LCMAQM has been applied in the U.S as well. In 2017 in Seattle, WA a low-cost sensor array was installed in a hybrid-electric vehicle to collect data of air quality (Larson et al., 2017). In San Francisco, CA, sensor arrays were mounted on two Google Street View mapping vehicles (Apte et al., 2017).



In this study we extend the range and application of LCMAQM by developing a wireless Mobile Autonomous Air Quality Sensor box (MAAQSbox) to measure air pollution. The MAAQSbox holds LCMAQM sensors (gas and particle) and a wireless broadcasting system, all available on the market. The gas sensors measure atmospheric concentrations of carbon monoxide (CO), ground-level ozone ($O_3$), nitrogen dioxide ($NO_2$), and nitric oxide (NO). The particle sensor is an Optical Particle Monitor (OPC) that measures $PM_1$, $PM_{2.5}$, and PM10. There are also sulfur dioxide ($SO_2$), lung deposited surface area (LDSA), and volatile organic compounds (VOCs) sensors In the MAAQSbox but they are not included in this assessment. The MAAQSbox is a tool to provide spatiotemporally resolved air quality information, helping to identify local hot-spots of air pollution. The data it provides can be processed to determine impact of urban design (street canyons), road management (stoplight timing), and vehicle technologies (emission indexes of light duty and heavy-duty vehicles). The MAAQSbox allows for autonomous measurement in host vehicles, as a mobile monitoring platform that has the unique capabilities of measurements in a variety of environments. A special feature of the MAAQSbox is protection from potentially damaging conditions, e.g. high humidity, rain, or car washes.

While autonomous LCMAQM operation allows for greater system flexibility, the applicability of the device is determined by the accuracy of the resulting measurement data. (Castell et al., 2017). This study examines the quality of the LCMAQM data by assessing the performance of MAAQSbox relative to Minnesota Pollution Control Agency AMS regulatory equipment. We conducted a calibration in the field evaluating the impact of temperature, relative humidity, and cross-sensitivity in the calculation of the concentrations.

## 2. Methods

### 2.1 Description of MAAQSbox

The MAAQSbox is a unique autonomous device that houses several gas and particle sensors, includes hazard protection and thermal conditioning of sample streams, and continuously broadcasts measured concentrations and system status. The MAAQSbox can function under various and extreme weather conditions. For instance, it has a system to protect the sensors during rain or relatively high humidity. It also works well in an extreme temperature range (-30°-30°). This is a key variable during winter in Minneapolis-Saint Paul metro area, MN where the MAAQSbox will be tested.

As shown in Fig.1, the MAAQSbox contains the sensors for detection, pumps for advecting flow through the device, a heater to control the temperature, and an Arduino for system control. The Arduino controls the pumps, valves, and other devices that protect the MAAQSbox from water and humidity, as outlined in Fig. 2. There is also a board with a SIM card for wireless data transmission. The MAAQSbox is designed to be mounted on a bus, specifically it will be located above the bus's driver seat, and it will sample the ambient environment ahead of the bus by pumping outside air from above the driver seat, near the front-loading door. The MAAQSbox could be also placed in a car, train, or other mobile platforms. The dimensions of the MAAQSbox are 99 cm length, 11 cm width, and 15 cm height. As shown in Fig.1, air flows through an impactor, removing particles larger than 10 μm. Then, a humidity sensor determines the relative humidity, giving a signal to the Arduino allowing



for determination of whether it is safe to advect flow into the sensor array. Subsequently, the sample air flows through a 3-way valve and heater to maintain temperature and humidity levels within desired targets. Finally, a 3-way connector distributes the air to sensor areas.

The MAAQSbox holds seven gas sensors and two particle sensors. The gas sensors included in the calibration are CO, $O_3$, $NO_2$, and NO (B4 sensors AlphaSense, Inc.) and particle that measures $PM_1$, $PM_{2.5}$, and $PM_{10}$ (OPC-N2 AlphaSense Inc.). These sensors were installed in a Flow Sensing Cell Apparatus (FSCA). The FSCA was designed considering the sensor dimensions and the flow required for low residence time that is < 10 s.

The environmental sensors include a humidity sensor (HX71-VI, Omega), an optical rain sensor that detects increased the

refraction of light on the windshield due to the presence of water, and the water sensor that detects the amount of water collected in a water trap associated with the impactor as shown in Fig. 1.

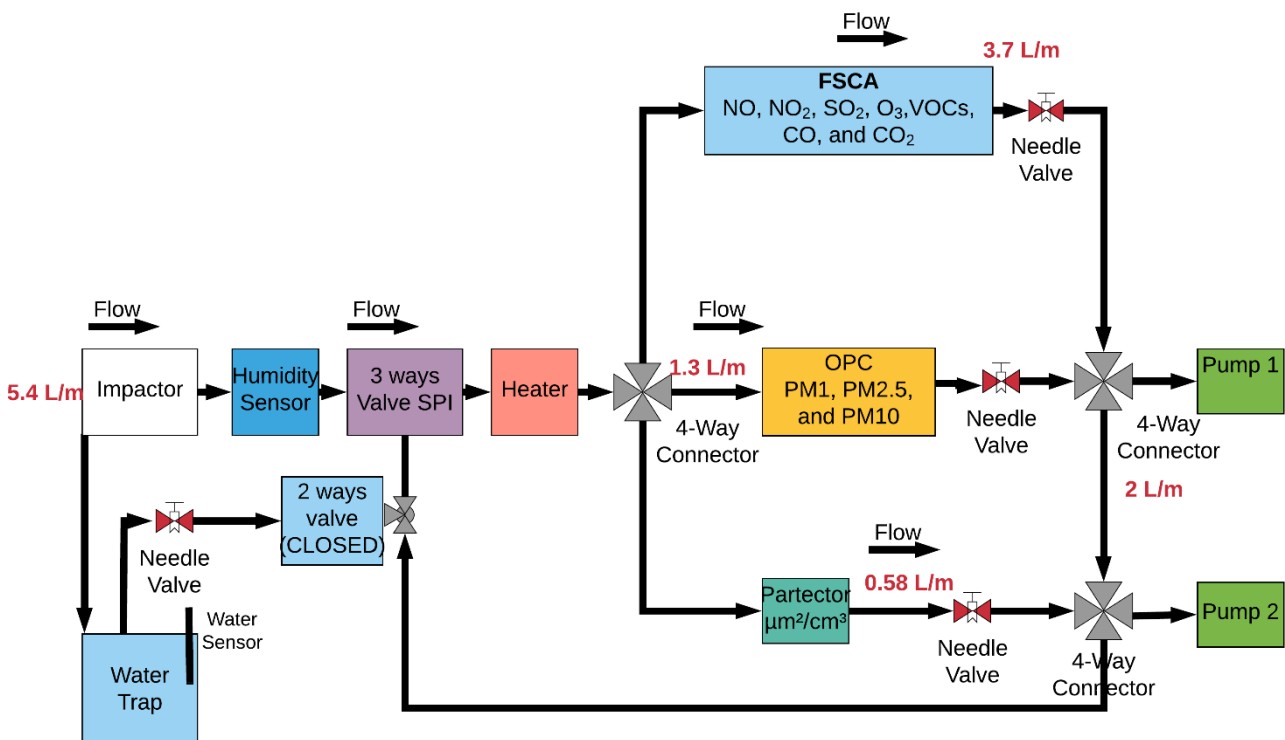

**Figure 1:** Schematic of The Mobile Autonomous Air Quality Sensor box (MAAQSbox). The pump vacuums the air from the Impactor to the whole system.




The heater system allows the MAAQSbox to maintain the temperature in the sensor measurement area at 28°C. The heater was developed to withstand winter in Minneapolis-Saint Paul metro area because low temperatures can affect sensor
performance and high relative humidity levels can damage the sensors.

The Arduino is an open-source platform used for building electronic projects. In the MAAQSbox, Arduino controls the pumps and valves and uses data from the humidity sensor, the water sensor, the temperature sensor, and the rain sensor. The Arduino prevents rainwater or high humidity from entering the system in two ways. When the rainwater sensor detects droplets, the pumps are turned off and the valves are switched to bypass the sensors. Similarly, if the relative humidity is equal or higher
than 80%, the valves switch to bypass the sensors. Fig.2 shows the support process using Arduino.

**Fig.2** Control logic to respond to rainwater and humidity in the MAAQSbox. The Arduino receives signals from the humidity and rain sensors, which it uses in a control platform to modulate the valves and pumps such that sensors are protected from damaging high humidity environments.






The signal of each sensor is processed by a Yocto-Board. The CO, O₃, NO₂, and NO, sensors use Yocto-0-10V-Rx boards. This board reads the instantaneous (1 Hz) value of any sensor following the 0-10 V standard (Yoctopuce, 2018a). The outputs from OPC-N2 are read by Yocto-SPI. The humidity, rain, and water sensors and valve position are read by Arduino. All data from Arduino is sent to Yocto-Serial. All the sensors and Arduino data are broadcast by YoctoHub-GSM-3G. The YoctoHub-

GSM-3G is a wireless-enabled module to access Yocto-0-10V-Rx, Yocto-RS232, Yocto-SPI, and Yocto-Serial remotely through a 3G GSM cellular network (Yoctopuce, 2018b). However, the data were retrieved by each Yocto board and Python instead of YoctoHub-GSM-3G. The sensors were set to collect data at 1 Hz.

## 2.2 Sensor Technology

### 2.2.1 NO₂, NO, CO, and O₃ Sensors

The NO₂, NO, CO, and O₃ gases are measured by AlphaSense B4 sensors. A B4 sensor contains three main components. As shown in Fig.3, from the top to the bottom, the first component is a gas chamber and a filter to improve gas selectivity. The second is an electrochemical cell where four electrodes are in a liquid electrolyte solution. Finally, in the lower section there is a reservoir of electrolyte solution and connections to the electrodes (Baron and Saffell, 2017). AlphaSense B4 sensors are electrochemical cells that generate a current that is linearly proportional to the fractional volume of the target gas species. Each

sensor contains Working, Auxiliary, Reference, and Counter electrodes (Spinelle et al., 2015). The target gas diffuses through a membrane where electrochemical oxidation (NO and CO) or reduction (NO₂ and O₃) occurs at the working electrode, generating a current signal (Mijiling et al., 2017; Spinelle et al., 2015). This electric signal is balanced by the counter electrode (AlphaSense, 2018; Mijiling et al., 2017). The reference electrode anchors the working electrode and helps to maintain working electrode performances and its sensitivity (AlphaSense, 2018). The auxiliary electrode is not exposed to the target gas. This is

to provide the background current to the current observed in the working electrode (Baron and Saffell, 2017). An individual sensor board also designed by AlphaSense was used to reduce environment noise achieving reported ppm or ppb resolution in accordance with the sensor specifications (AlphaSense, 2016; Mijiling et al., 2017).



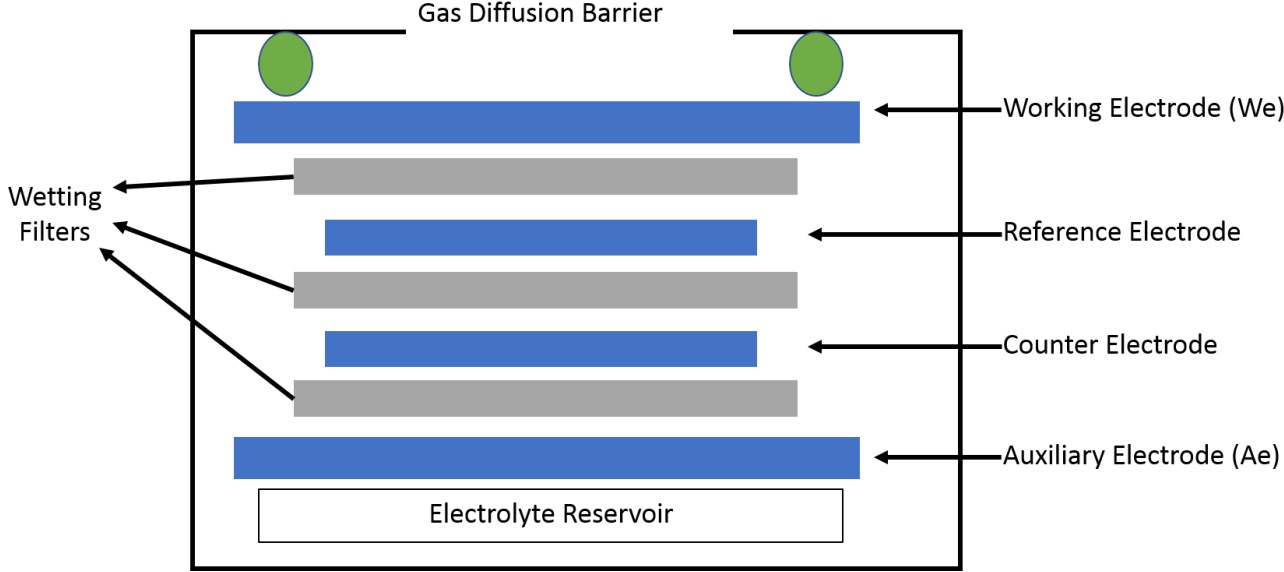

**Fig. 3.** Plot a Schematic diagram of electrochemical toxic gas sensor. From the top to the bottom: Working, Auxiliary, Reference, and Counter electrodes.

### 2.2.2 Particle Sensor: Optical Particle Monitor (OPC-N2)

OPC-N2 is a particle sensor developed by AlphaSense. OPC-N2 obtains measurements by passing particles through a chamber illuminated by a laser beam scatter light into the particle optical detector (Walser et al., 2017; AlphaSense, 2015). The intensity of light scattered from aerosol particles is a function of particle size, shape, and composition. The OPC-N2 measures size and number of particles in 16 bins from 0.38 µm to 17 µm (Sousan et al., 2016). The mass concentrations for three particle sizes, $PM_1$, $PM_{2.5}$, and $PM_{10}$, are calculated using the measurements of size and number of particles (AlphaSense, 2015). The OPC-N2 assumes spherical particles with a density is 1,650 kg m-3 and a volume-weighting factor of 1.0 (Sousan et al., 2016).

### 2.3 Calibration

The calibration process includes two processes: laboratory calibration and field calibration.

### 2.3.1 Calibration in the laboratory

The calibration in the lab was conducted for CO, NO, and $NO_2$. The references were cylinders with certain concentrations. The concentrations were calculated using the equation provide by the manufacturer. This equation does include the temperature but not the relative humidity. The Eq. (1) presents details of the variables included.



$$Conc\,[\text{ppb or ppm}] = ((We - We_{zero}) - (Ae - Ae_{zero}) \times T_{Correct})/Sens, \qquad (1)$$

Where $We$ is working electrode voltage, $We_{zero}$ is working electrode offset error, Ae is auxiliary electrode, Aezero is auxiliary

electrode offset error, $T_{Correct}$ is Temperature correction, and $Sens$ is the sensitivity [V/ppm or V/ppb].

The working electrode offset error, auxiliary electrode offset error and sensitivity were obtained from best fits to data. The

temperature factors were provided by the manufacturer. The correlation between reference and measured concentrations were

0.99, 0.98, and 0.95 for CO, NO, $NO_2$, respectively. There were no standard calibration generators of $O_3$ or particles available

for the respective ozone and particle sensors. When the calibration constants obtained in the laboratory were applied for outside

measurements, there were reported instances of negative concentrations, as well as significant differences between LCMAQM

and air monitoring station (AMS) references. Based on these unsatisfactory results we decided to perform a field calibration

using methods similar to those reported by other investigators (Castell et al., 2017; Cross et al., 2018; Spinelle et al., 2015;

Walser et al., 2017; Hagan et al., 2018; Mijiling et al., 2017; Zimmerman et al., 2017).

**2.3.2 Calibration in the Field**

The aim of the field calibration is to evaluate the low-cost sensor performance compared to a reference instrument in the field.

An AMS provides robust data of the air pollution. Thus, the low-cost sensors were operated side by side with an AMS. This

calibration is a key step to obtain reliable data because it includes actual environmental conditions and air quality data for an

extended period, typically a few weeks. The calibration does not aim to achieve conclusions about the air quality, only

calibration. The sensors provide valuable data but require careful calibration not only in pre-operation stage but also

periodically throughout operation (Castell et al., 2017; Cross et al., 2018; Spinelle et al., 2015; Walser et al., 2017; Hagan et

al., 2018; Mijiling et al., 2017; Zimmerman et al., 2017).

The calibration in the field was conducted during September 2018 with the assistance of the Minnesota Pollution Agency

(MPCA). The MAAQSbox was placed next to a MPCA AMS in Minneapolis in a near-road location just south of Interstate

94. This AMS provides air pollutant concentrations for $PM_{2.5}$, $O_3$, CO, NO, and $NO_2$. The sensors in the AMS are Teledyne

T200 $NO_x$ for NO and $NO_2$, T300 for CO and T400 for Ozone. The BAM 1020 (Metone) measures $PM_{2.5}$ by measuring the

attenuation of a beam of beta particles passing through particulate matter collected on a filter tape.

The inlet to the MAAQSbox inlet was at the same height and facing the same direction as the AMS inlet. The horizontal

difference between the MAAQSbox and AMS inlets was ~30 cm. The field calibration was conducted for 154 hours. Data

from the AMS are reported as one-hour averages so MAAQSbox data were averaged over the corresponding periods giving

154 one-hour averages. Fig. 4(a) shows the inlet of AMS and MAAQSbox while 4(b) is a front view of the AMS. The AMS

is represented by the red circle in Fig. 4(c).





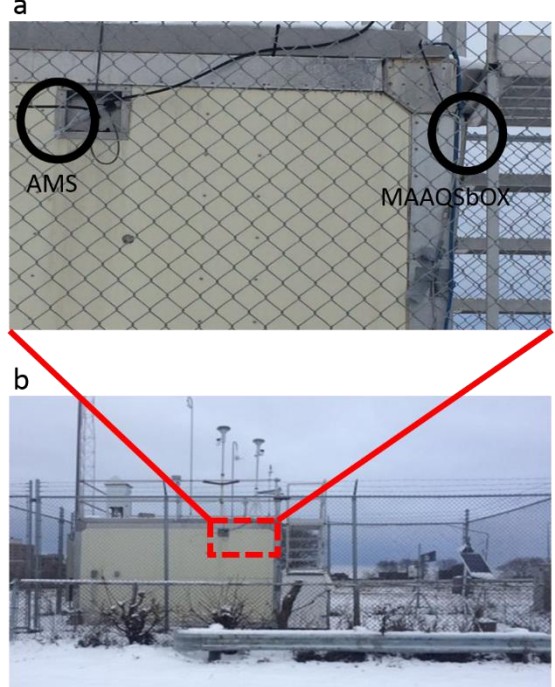

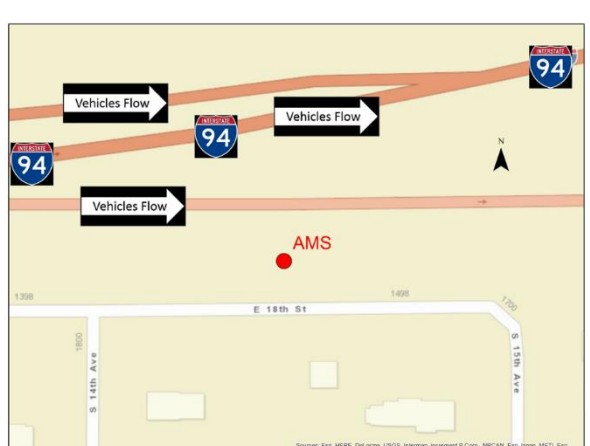


**Fig. 4.** The inlets of the Air Monitoring Station (AMS) and the Mobile Autonomous Air Quality Sensor box (MAAQSbox) (a). Zoomed view of the Near-Road AMS (b). Near-Road AMS. (c)The location of the AMS in Minneapolis. © OpenStreetMap contributors 2019. Distributed under a Creative Commons BY-SA License.

The calibration of LCMAQM sensors were determined by multivariate linear regressions (MLR); the independent variables were signals and data from LCMAQM and environmental sensors, while the dependent variable was the concentration of air pollution from AMS. The independent variables of MLR for LCMAQM gas sensors ($NO_2$, NO, CO, and $O_3$) were We and Ae sensor signals. Also, temperature and humidity in the MAAQSbox were included as independent variables in MLR.

   The particle sensor is handled differently. The OPC-N2 provides the air pollution concentration in microgram per cubic meter

($\mu$g m$^{-3}$) as a factory calibration; however, MLR was still performed to calibrate the $PM_{2.5}$ concentration. The MLR includes as independent variables the OPC-N2 data, temperature, and humidity. The OPC-N2 calibration was conducted with the AMS particle sensor as a reference.

   We assumed the relationship between independent variables and the dependent variable is linear. This assumption is justified for three reasons. First, the scatter plot of data suggests a straight-line model. Second, MLRs were used in other projects using

the same sensor technologies (Castell et al., 2017; Cross et al., 2018; Hagan et al., 2018). Third, the linear relationship between variables is generally the most parsimonious (Lewis-Beck, 1980). The coefficient of determination ($R^2$) and the adjusted coefficient of determination (adjusted-$R^2$) and p-value have been evaluated to determine the best-fit model. $R^2$ is a biased





estimator that increases as independent variables are added to the model, the adjusted-$R^2$ reduces the bias (Ricci, 2010), allowing different models to be compared. The range of p-value is between zero and one. A low p-value ($<0.05$) indicates that

a specific independent variable is meaningful for the best-fitted model. However, it does not provide information about the magnitude of the impact of a specific variable (Goodman, 2008).

The root-mean-square error (RMSE) is the root-mean of the square of the difference between the observed value (data from AMS) and the predicted value. The RMSE is a measurement of goodness of fit, lower RMSE means lower error or better fit of MLR. The regressions were performed in Matlab by minimizing the sum of the squared errors. The procedure to determine

the variables that is included in the best-fitted model for gas sensors is the following:

1. Perform a regression with the working electrode (We) and the auxiliary electrode (Ae) sensor signals.

2. Perform a regression with the We and the Ae but including the data from temperature and humidity sensors.

3. Compare the Adjusted-R2 of the both regressions.

4. Evaluate the p-value of the temperature and humidity sensor data.

5. If the p-value is $>0.05$ and the Adjusted-R2 did not increase, the variable will be removed from the model, performing another regression.

There is a known cross-sensitivity of $O_3$ and $NO_2$ with the AlphaSense sensors (Spinelle et al., 2015; Mijiling et al., 2017; manufacturer guidelines) so $O_3$ was included in the $NO_2$ calibration and $NO_2$ in the $O_3$ calibration. For PM2.5 best-fitted model, instead of the *We* and *Ae*, the concentration in µg m$^{-3}$ provided by the sensor is used.

## 3. Results

### 3.1 Models

The Eqs. (2-6) listed below represent the best-fitted model for each sensor. CO and NO sensor models include the working and auxiliary electrodes, temperature, and humidity. The $NO_2$ and $O_3$ contain the working and auxiliary electrodes, temperature, and humidity, but also working and auxiliary electrodes of other sensors including the cross-sensitivity effect.

The OPC-N2 contains the sensor signal and temperature. The details of each model and the independent variables included are presented below.

$$CO\,[\text{ppm}] = \beta_0 + \beta_1 *We_{COsensor} + \beta_2 \times Ae_{COsensor} + \beta_3 \times T° + \beta_4 \times RH \tag{2}$$

$$NO\,[\text{ppb}] = \beta_0 + \beta_1 *We_{NOsensor} + \beta_2 \times Ae_{NOsensor} + \beta_3 \times T° + \beta_4 \times RH \tag{3}$$

$$NO_2\,[\text{ppb}] = \beta_0 + \beta_1 *We_{NO2sensor} + \beta_2 \times Ae_{NO2sensor} + \beta_3 \times We_{O3sensor} + \beta_4 \times Ae_{O3sensor} + \beta_5 \times T° + \beta_6 \times RH \tag{4}$$

$$O_3\,[\text{ppb}] = \beta_0 + \beta_1 \times We_{O3sensor} + \beta_2 \times Ae_{O3sensor} + \beta_3 \times We_{NO2sensor} + \beta_4 \times Ae_{NO2sensor} + \beta_5 \times T° + \beta_6 \times RH \tag{5}$$

$$PM_{2.5}\,[\text{ug m}^{-3}] = \beta_0 + \beta_1 \times OPC\text{-}N2 + \beta_2 \times T° \tag{6}$$



Where $B_0$ is the intercept when all independent variables are zero, $B_x$ is the regression coefficient, *We* is working electrode of
x sensor, *Ae* is auxiliary electrode of x sensor, $T°$ is temperature in Celsius, and *RH* is relative humidity.

As shown in Table 1, the gas sensors correlate well measured data from the AMS, with R2 > 0.9 for all but NO$_2$ which has $R^2$
= 0.81. The fit improved for all gas sensors when the temperature and humidity were included, lowering RMSE in all cases.
The largest effect in RMSE due to inclusion of the temperature and humidity was in NO sensor, which was reduced from
RMSE = 8.1 to 3.4.


**Table 1.** Calibration results of B4 gas sensors and OPC-N2. The first column is the sensor air pollution; the second column is the number of
samples in hours; the third is the coefficient of determination ($R^2$); the fourth is the root-mean-square error; the fifth and sixth columns are
the $R^2$ adjusted without and with temperature and humidity respectively; and seventh is the average of the measurements of AMS.

**Table 1**

| Sensor | N | $R^2$ | RMSE | $R^2$ Adj. | $R^2$ Adj. w/ $T°$ and RH | Average |
|--------|---|-------|------|-----------|--------------------------|---------|
| CO | 154 | 0.957 | 0.0531 | 0.946 | 0.956 | 0.37 ppm |
| NO | 154 | 0.972 | 3.4 | 0.83 | 0.971 | 19.5 ppb |
| NO$_2$ | 154 | 0.812 | 3.09 | 0.742 | 0.804 | 14.6 ppb |
| O$_3$ | 154 | 0.947 | 2.24 | 0.928 | 0.945 | 12.4 ppb |
| PM$_{2.5}$ | 154 | 0.599 | 6.18 | 0.542 | 0.593 | 5.2 ug m$^{-3}$ |

The adjusted-$R^2$ increased from 0.946 to 0.956 for CO; from 0.835 to 0.971 for NO; from 0.742 to 0.804 for NO$_2$; from 0.928
to 0.945 for O$_3$. Including the temperature and humidity data all the gas sensor models improved their capacity to predict the
concentrations. The p-values of the temperature and humidity data for all gas sensor models are lower than 0.05. Thus, we can
conclude that temperature and humidity are significant variables in the process of calculate the concentrations.

The PM$_{2.5}$ sensor had varied performance with an $R^2$ = 0.599. The PM$_{2.5}$ increased from 0.542 to 0.593 but only including
temperature as new independent variable. The relative humidity was not included in the model because the p-value is higher
than 0.05.

Figs. 5(a-e) show the results of low-cost sensors and reference sensors for each species tested. The abscissa represents the time
of measurements and ordinate represents the concentration of pollutant. In CO (Fig. 5(a)) the results present similar trends.
The mean of the differences between AMS and MAAQSbox is 0.038 ppm for a mean CO concentration of 0.36 ppm. However,





there are differences of ~0.1 ppm during a few hours (12 data points) of the calibration, specifically at AMS concentrations

higher than 0.5 ppm. The one-hour average standard of EPA is 35 ppm; thus, the CO sensor measurements here were at concentrations far below the EPA standard, detecting reliably at concentrations down to 0.1 ppm. The NO (Fig. 5(b)) sensor and references also have a similar trend. There are twenty hourly averages where there were differences between ~5 ppb and 11 ppb at concentrations of ~1 ppb and concentrations higher than 30 ppb. This represents 13% of the sample. The $NO_2$ (Fig. 5(c)) is shown to have differences between ~5 ppb and 6 ppb. These differences are found at low, high, and average

concentrations. The mean of measured concentrations over our study window was 14.6 ppb while one-hour average EPA standard is 100 ppb. The $O_3$ sensor and reference presents a similar trend. As is shown in Fig. 5(d), the $O_3$ sensor is accurate in all levels of measured concentrations. The average of concentrations is 12.4 ppb while eight-hour average standard of EPA is 70 ppb. The $PM_{2.5}$ average of AMS measurements is 5.2 µg m$^{-3}$ and average of the differences between AMS and MAAQSbox is 3 µg m$^{-3}$. The differences higher than 5 µg m$^{-3}$ are found in AMS concentrations higher than 9 µg m$^{-3}$ and

lower than 5 µg m$^{-3}$. It is important to mention that 8% of the data from AMS are negative concentrations suggesting possible issues with this instrument. As shown in Fig. 5 (e) the OPC-N2 does not track well the trends in the first fifty hours. During these hours OPC-NO2 does not track the peaks and lowest concentrations.

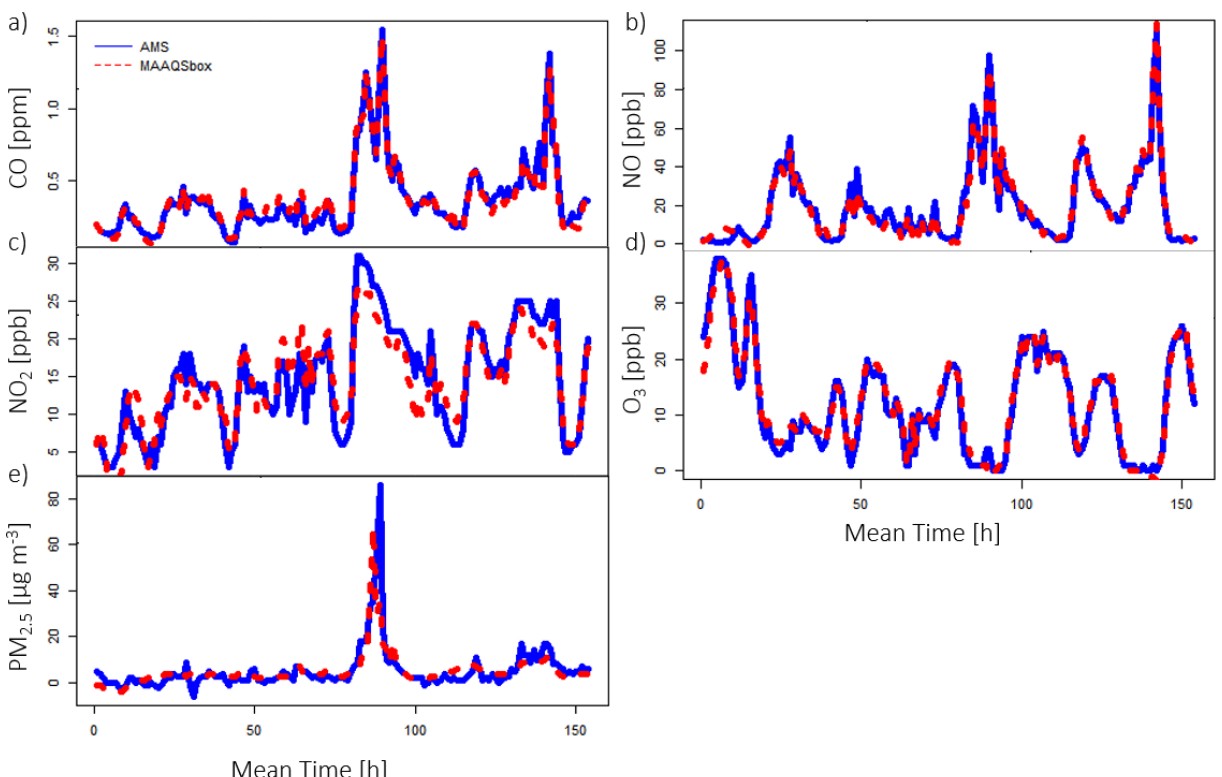

**Fig. 5.** Mean hourly concentration of pollutants (a) CO, (b) NO, (c) NO₂, (d) O₃, and (e) PM2.5 for measured AMS and MAAQSbox data

over 154 hours period



Figs. 6(a-e) are scatterplots where the ordinate is data from sensors (MAAQSbox) while abscissa represents the data from reference (AMS). CO (Fig. 6 (a)), NO (Fig. 6 (b)), and $O_3$ (Fig. 6 (d)) present higher slope, which means better fit to the data

from AMS than the data for $NO_2$ and OPC-N2. The slopes are 0.99, 0.98, and 0.95 for CO, NO, and $O_3$ sensor respectively. The Fig. 6(c) shows the data for $NO_2$. There is more scatter in the data compared to that from CO, NO, and $O_3$. The slope of $NO_2$ is 0.79. The OPC-N2 data are presented in Fig. 6(e), which has the worst comparison between AMS and MAAQSbox data with a fit slope of 0.59. There are several outliers, especially in high concentrations. Removing the four highest concentrations (boxed) the slope improves to 0.71.




**Fig. 6.** Regression of pollutant concentrations for (a) CO, (b) NO, (c) NO$_2$, (d) O$_3$, and (e) PM$_{2.5}$ MAAQSbox relative to the AMS reference
monitors. The equation for each linear regression is provided.

## 4. Conclusions

We evaluated low-cost CO, NO, NO$_2$, O$_3$ B4 sensors, and OPC-N2 particle sensor, all of them produced by AlphaSense. The
CO, NO, and O$_3$ sensors presented a high R$^2$ (> 0.9), which shows good agreement with the fitted regression line. These values
are similar compared to other projects using the same technology. The NO$_2$ sensor presents an R$^2$ = 0.81, which is below that
of the other B4 sensors. Poorer performances for NO$_2$ has also been reported by other researchers where R$^2$ = 0.4to 0.89
(Castell et al., 2017; Mijiling et al., 2017). The OPC-N2 shows an R$^2$ = 0.599. This result also is similar to other experiences
using OPC-N2.

The commercially provided sensor B4 sensitivity and equation to convert outputs from volts to relevant units of pollution
concentration (ppm or ppb) did not provide agreement between measured and reference values. Use of the default manufacturer
relations resulted in non-physical negative concentrations and larger gap between MAAQSbox and AMS data. In this context,
building a model using MLR reduce the gap between sensors and AMS results. The temperature and relative humidity increase
the R$^2$ of the all models, thus they must be included in the calculation of concentrations. The calibration in the field is highly
recommended before conducting any measurements with these low-cost sensors, which have been shown to be affected by
temperature and relative humidity. The calibration must be conducted periodically because the sensitivity of sensor changes
over time, which we anticipate to be ~3 months. With appropriate calibration and monitoring of sensor performance we
conclude that the MAAQSbox sensing platform can achieve reliable ambient air quality measurements for most of the
pollutants examined.

**Data availability**

The data from Air Monitoring Station (AMS) are available in AirNow  www.airnowtech.org. The data of the Low-cost sensors
are available from Andres Gonzalez (gonza817@umn.edu)

**Author contributions**

AA, AB, and JS designed the MAAQSbox. The calibration in the field was designed by AA and DK and conducted by AA.
AA performed the data analysis while the manuscript was prepared with contributions from all co-authors.

**Competing interests**





The authors declare that they have no conflict of interest.

**Disclaimer**

Reference herein to any specific commercial product, process, or service by trade name, trademark, manufacturer, or otherwise does not constitute or imply its endorsement or recommendation by the authors.

**Acknowledgments**

This research is supported by University of Minnesota Institute on the Environment. We thank our colleagues from the Department of Civil, Environmental, and Geo- Engineering and the Department of Mechanical Engineering, specially Mugurel Turos and Bernard Olson. We would like also thank Minnesota Pollution Control Agency for allowing us to install our MAAQSbox in their AMS and Matt Robertson (MSU Mankato student) for assisting in the design and fabrication of the flow cell apparatus.

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
