# Peer review of "Field Calibration of Low-Cost Air Pollution Sensors"

_Atmospheric Measurement Techniques, 2019_

## Referee Comment (RC1) · Anonymous Referee #1 · 6 Sep 2019

The authors have constructed a relatively compact, 'low cost', transportable air pollution monitoring system (which they call a Mobile Autonomous Air Quality Sensor box – MAAQSbox) that contains a suite of commercially-available AlphaSense B4 electrochemical sensors for CO, NO, NO2 and O3, and an AlphaSense OPC-N2 optical particle monitor for PM2.5 (and other PM size fractions). The MAAQSbox also includes thermal conditioning of the inlet sampling stream, which incorporates internal T and RH sensing, and wireless data transmission.

The purpose of this paper is to report the results of a field calibration of the above-listed air pollutant sensors in the MAAQSbox as derived from co-location of the box at a 'reference' Air Monitoring Station in a 'near-road' location in Minneapolis, USA. (The paper states that the MAAQSbox also contains sensors for SO2, CO2 and VOC, but

only calibration for the five pollutant sensors mentioned above is reported here.)

There is considerable motivation for developing lower-cost air pollution measurement instrumentation; namely, the large adverse health impacts caused by poor air quality and the consequent need to collect more spatially spread measurements in order to better characterise human exposure, to develop and evaluate models and, subsequently, to evaluate effectiveness of mitigation. Description of some of the above motivation is appropriately covered by the authors in the Introduction to their paper.

The authors have clearly given good thought to construction of their MAAQSbox, particularly the efforts to provide some T and RH conditioning of the incoming air stream in order to negate dependencies of sensor response to these ambient variables.

In contrast, however, the calibration work described here is extremely limited, and, in my view, not of sufficient level to support full publication of this work in AMT. In both the Introduction (Line 80) and the Methods (Line 180) it is stated that the aim of this work is to 'evaluate' the performance of the sensors in their MAAQSbox, yet the design of their study and the data presented only reports a calibration (and not even the full calibration data), not an evaluation.

The co-location dataset comprises a single 6.5-day (154 hour) co-location at one time of year at one site. This dataset is then used to derive a multivariate linear regression calibration for each sensor against its relevant 'reference instrument' value using sensor signals and the internal airstream T and RH values as dependent variables. For calibration of the NO2 and O3 sensors, sensor signals from the other species sensor were also included to allow for potential cross-species interference. However, the authors do not present the actual calibration equation coefficient values and their p values (they only state which variables are included in each sensor calibration equation). Nor do they present visualisations and/or statistics for the raw comparisons of sensor values against respective reference concentrations. Consequently, in the absence of such information the reader is not able to gauge how well or not each sensor performs

prior to the multivariate regression fits, i.e. to gauge how much modification to raw sensor output is being made by the derived multiple regression calibration equation. In other words the reader does not get a sense of how much the sensor signal needs to be corrected for the contribution of other variables to the signal, particularly the extent to which there has to be correction for cross-interference between the NO2 and O3 sensors. Such information would tell the reader how important other variables are.

A more fundamental flaw, however, is that there is no independent evaluation of the calibration: the same data is used both to derive a calibration equation and then to justify the goodness of the calibration once applied to that data. If one derives a predictor equation from a dataset and then applies the predictor equation to exactly the same dataset then of course the predictions (and their 'evaluation' statistics) are likely to be very good. At the very least, there needs to be sufficient co-location data to (randomly) split into 'training' and 'test' sub-datasets in order to provide some (quasi)independent statistical evaluation of a derived calibration. More usefully still, what potential users of this MAAQSbox need to know is how well does a calibration equation hold in time and at different locations. Is there evidence of any long-term drift in sensor performance/calibration? If the sensors in the MAAQbox is calibrated at one location, does the same calibration hold at another location and/or at another time? If the MAAQbox is calibrated prior to a mobile deployment and is then used as intended on a mobile platform how well does its calibration hold up when the MAAQBox is co-located back at the reference monitoring station? All that the data presented in this paper show is that an underlying relationship for sensor performance self-consistently holds within a single 154 hour period.

Some additional comments:

The regression equation written on the panel of Figure 6c does not seem correct. The intercept appears to be much larger than 0.29 ppb, and eyeballing this panel suggests that the plotted regression line is giving much higher values for estimated NO2 than the stated regression line would predict; for example, for a reference value of 15 ppb

the regression equation predicts a sensor value of 12.14 ppb but the plotted line shows higher estimated NO2 than this. Also, this panel should include the origin of the scatter plot.

The scatter plot in panel 6a should also include the origin of the plot, and why does the regression equation for this panel not have an intercept coefficient? Even if the coefficient is not statistically significant its value should be included to indicate that the regression included an intercept in the fit.

The increased scatter in the calibration scatter plot for NO2 (Figure 6c) is noted but there is no discussion of this. Given that NO2 is a key pollutant in the urban environment, for which quantification by instruments such as MAAQSbox is most keenly sought, there needs to be further comment on what is underlying this poorer performance for NO2 measurement. As indicated above, we are not given the magnitudes or p values of coefficients in the calibration equations: which one of the variables is having the most influence on the NO2 response during this deployment?

In places the paper contains too much description of material that the authors can assume readers of this paper will be aware of. For example, it is not necessary to provide so much description of what statistical metrics such as R2, RMSE and p values mean. There is also quite a lot of generic description of operation of an electrochemical gas sensor.

---

## Short Comment (SC1) · 23 Sep 2019

The paper states that the MAAQSbox also contains sensors for SO2, CO2 and VOC, but only calibration for the five pollutant sensors mentioned above is reported here

• The Air Monitoring Station using as reference does not measure SO2, CO2 and VOC. • Generally, CO2 is not measure in air monitoring station. • The concentration of SO2 is $\sim$ 0 ppb in the other air monitoring stations therefore we are not going to include in the measurements. • The data of VOC is limited in other air monitoring stations. The VOC measurement was also discarded.

In both the Introduction (Line 80) and the Methods (Line 180) it is stated that the aim of this work is to 'evaluate' the performance of the sensors in their MAAQSbox, yet the

design of their study and the data presented only reports a calibration (and not even the full calibration data), not an evaluation. • This is correct, we should not have used the word evaluation but examine low-cost sensor previous short term (2 months) measurements. • Also, we can write more about our results and explain/show other results obtained during calibration.

The co-location dataset comprises a single 6.5-day (154 hour) co-location at one time of year at one site. This dataset is then used to derive a multivariate linear regression calibration for each sensor against its relevant 'reference instrument' value using sensor signals and the internal airstream T and RH values as dependent variables. For calibration of the NO2 and O3 sensors, sensor signals from the other species sensor were also included to allow for potential cross-species interference.

• We only show 156 hours. However, there are other 244 for CO, 170 for NO, and 87 for NO2 and O3 hours of calibration running 5 months later with the same sensors.

However, the authors do not present the actual calibration equation coefficient values and their p values (they only state which variables are included in each sensor calibration equation). • we can add that data

Nor do they present visualisations and/or statistics for the raw comparisons of sensor values against respective reference concentrations. Consequently, in the absence of such information the reader is not able to gauge how well or not each sensor performs prior to the multivariate regression fits, i.e. to gauge how much modification to raw sensor output is being made by the derived multiple regression calibration equation. In other words the reader does not get a sense of how much the sensor signal needs to be corrected for the contribution of other variables to the signal, particularly the extent to which there has to be correction for cross-interference between the NO2 and O3 sensors. Such information would tell the reader how important other variables are. • we can add that data

A more fundamental flaw, however, is that there is no independent evaluation of the

calibration: the same data is used both to derive a calibration equation and then to jus-tify the goodness of the calibration once applied to that data. If one derives a predictor equation from a dataset and then applies the predictor equation to exactly the same dataset then of course the predictions (and their 'evaluation' statistics) are likely to be very good. At the very least, there needs to be sufficient co-location data to (randomly) split into 'training' and 'test' sub-datasets in order to provide some (quasi)independent statistical evaluation of a derived calibration. More usefully still, what potential users of this MAAQSbox need to know is how well does a calibration equation hold in time and at different locations. Is there evidence of any long-term drift in sensor performance/ calibration? • There is evidence of drift in long-term sensor. We can show the two different calibrations conducted by 5 months away. • We can apply the equation if the data presented on this paper to the second calibration and assess the change in raw data and see potential drift.

If the sensors in the MAAQbox is calibrated at one location, does the same calibration hold at another location and/or at another time? If the MAAQbox is calibrated prior to a mobile deployment and is then used as intended on a mobile platform how well does its calibration hold up when the MAAQBox is co-located back at the reference monitoring station? • This was answered above.

All that the data presented in this paper show is that an underlying relationship for sensor performance self-consistently holds within a single 154 hour period. • This was answered above.

Some additional comments: The regression equation written on the panel of Figure 6c does not seem correct. The intercept appears to be much larger than 0.29 ppb, and eyeballing this panel suggests that the plotted regression line is giving much higher values for estimated NO2 than the stated regression line would predict; for example, for a reference value of 15 ppb the regression equation predicts a sensor value of 12.14 ppb but the plotted line shows higher estimated NO2 than this. Also, this panel should include the origin of the scatter plot. • We are going to double check this chart and

the calculation. We will include any change due to new calculations. As soon as we find out the error, we will do a clarification. • In this case the reviewer was correct. The equation should be: y = 0.7873x + 2.8665 • We are going to include the origin of the scatter plot.

The scatter plot in panel 6a should also include the origin of the plot, and why does the regression equation for this panel not have an intercept coefficient? Even if the coefficient is not statistically significant its value should be included to indicate that the regression included an intercept in the fit. • We are going to double check this chart and the calculation. We will include any change due to new calculations. As soon as we find out the error, we will do a clarification. • In this case the reviewer was correct. The equation should be: y = 0.9569x + 0.0159 • We are going to include the intercept

The increased scatter in the calibration scatter plot for NO2 (Figure 6c) is noted but there is no discussion of this. Given that NO2 is a key pollutant in the urban environment, for which quantification by instruments such as MAAQSbox is most keenly sought, there needs to be further comment on what is underlying this poorer performance for NO2 measurement. • We can say more about it and describe/explain the NO2 sensor performance. We can include potential reasons for (poor) the NO2 performance

As indicated above, we are not given the magnitudes or p values of coefficients in the calibration equations: which one of the variables is having the most influence on the NO2 response during this deployment? • As we mentioned, we can include all the p value and coefficient of each variable of each sensor.

---

## Referee Comment (RC2) · Anonymous Referee #2 · 27 Sep 2019

This paper describes a custom monitoring device, the Mobile Autonomous Air Quality Sensor box (MAAQSbox), for the measurement of several important gas phase air pollutants and atmospheric particulate matter. The MAAQSbox is based on commercially available "low cost" electrochemical sensors for gases and a "low cost" optical particle counter for particle measurements. The paper describes the sensor technology as well as the custom sampling and control system. This includes temperature control of the sensor measurement area and an inlet to protect the sensors from rain and high humidity, via an inlet bypass, as well as measurements of air temperature and humidity. The authors also compare data from the MAAQSbox with reference grade instrumentation and use this to develop linear calibration equations. Overall this work could be a useful addition to the growing literature on custom "low cost" air pollution sensor devices, but

requires more evidence on the performance evaluation of the MAAQSbox and major improvements to the calibration section of the paper before it should be published in AMT.

Major comments:

1) The most significant flaw in the analysis presented is that it seems that the data used to train the calibration models are the same used to evaluate the same models? If so this is not a valid test, and the training and test data need to be independent data sets.

2) Overall the calibration approach is not clear, with no indication of the improvement achieved with the increasing complexity of the calibration equation used. It would be helpful to the reader if the authors could provide a baseline performance of the sensors using a simple linear fit to the raw sensor signals, before including other variables such as temperature. This would enable the impact of sensor interferences, e.g. from temperature, to be understood in both the laboratory and field calibrations.

3) As this is a description of a new instrument the authors should provide an assessment of the measurement uncertainty.

4) The poor performance seen for the OPC-N2 sensor when compared to reference measurements is not adequately discussed. Early studies using these sensors identified a significant humidity dependence impacting the data under high humidity conditions. A study by Antonio et al. (2018) developed a correction for this instrumental effect on the OPC-N2, resulting in an apparent improvement in data quality. The authors should at the very least acknowledge this earlier work and discuss the implications for the work presented here.

Minor comments:

1) Table 1 has no units on values other than the average mixing ratio.

2) The statement on line 321 that calibrations will last $\sim$ 3 months has no supporting evidence and should either be removed or justified.

References: Di Antonio A, Popoola OA, Ouyang B, Saffell J, Jones RL. Developing a relative humidity correction for low-cost sensors measuring ambient particulate matter. Sensors (Switzerland) 2018;18:2790. doi: 10.3390/s18092790.

———————————————————

---

## Short Comment (SC2) · 15 Oct 2019

1) The most significant flaw in the analysis presented is that it seems that the data used to train the calibration models are the same used to evaluate the same models? If so this is not a valid test, and the training and test data need to be independent data sets. The purpose of these tests is to determine how well field calibrations of the sensors works. It is being done in the presence of many unknown and uncontrolled ambient variables. This is different from a lab calibration where, for example you might compare instrument response to a pure span gas setting, with few uncontrolled variables. In these experiments we compare the response of the sensors to reference field sensors and determine how well a multiple linear regression model can relate the sensor response to the variable in question. 2) Overall the calibration approach is not clear,

with no indication of the improvement achieved with the increasing complexity of the calibration equation used. It would be helpful to the reader if the authors could provide a baseline performance of the sensors using a simple linear fit to the raw sensor signals, before including other variables such as temperature. This would enable the impact of sensor interferences, e.g. from temperature, to be understood in both the laboratory and field calibrations. We do add this information and analysis. Specifically showing the raw sensor signal compared to data from air monitoring station. 3) As this is a description of a new instrument the authors should provide an assessment of the measurement uncertainty. We do add the uncertainty of each sensor calibration not only from our calibration but also from literature review. 4) The poor performance seen for the OPC-N2 sensor when compared to reference measurements is not adequately discussed. Early studies using these sensors identified a significant humidity dependence impacting the data under high humidity conditions. A study by Antonio et al. (2018) developed a correction for this instrumental effect on the OPC-N2, resulting in an apparent improvement in data quality. The authors should at the very least acknowledge this earlier work and discuss the implications for the work presented here. We do add the implications of the others in term of OPC-N2 calibration assessing the impact in our results. 1) Table 1 has no units on values other than the average mixing ratio. We do add the unit of each value in Table 1. 2) The statement on line 321 that calibrations will last _ 3 months has no supporting evidence and should either be removed or justified. References: Di Antonio A, Popoola OA, Ouyang B, Saffell J, Jones RL. Developing a relative humidity correction for low-cost sensors measuring ambient particulate matter. Sensors (Switzerland) 2018;18:2790. doi: 10.3390/s18092790. We do add the reference(s) for the statement on line 321.

---

## Author Comment (AC1) · 15 Nov 2019

All the answers are the PDF document.

[Figure]

*(1). The paper states that the MAAQSbox also contains sensors for SO2, CO2 and VOC, but only calibration for the five pollutant sensors mentioned above is reported here*

(2)

- The air monitoring station using as reference does not measure $SO_2$, $CO_2$ and VOC.
- Generally, $CO_2$ is not measured in air monitoring station.
- The concentration of $SO_2$ is ~ 0 ppb therefore we are not going to include in the measurements.
- The data of VOC is limited to other air monitoring stations that don't measure CO, NO NO2 and O3 and PM2.5. The VOC calibration would have required a separate study.

(3).

[Figure]

"The MAAQSbox holds five gas sensors and two particle sensors. The gas sensors included in the calibration are CO, $CO_2$ $O_3$, $NO_2$, and NO (B4 sensors AlphaSense, Inc.)"
* * *
*(1). In both the Introduction (Line 80) and the Methods (Line 180) it is stated that the aim of this work is to 'evaluate' the performance of the sensors in their MAAQSbox, yet the design of their study and the data presented only reports a calibration (and not even the full calibration data), not an evaluation.*

(2)

- A new aim of this study was established including the reviewer comments for line 80 and 180.

(3)

Line 85. "We seek to relate sensor response, temperature, humidity, and concentrations other species exhibiting cross sensitivities to reference field measurements performs. Our analysis seeks to determine the accuracy and precision with which low-cost sensors perform during periodic "in-use" calibration."

**Fig. 1.**

---

## Author Comment (AC2) · 15 Nov 2019

(1) The paper states that the MAAQSbox also contains sensors for SO2, CO2 and VOC, but only calibration for the five pollutant sensors mentioned above is reported here

(2) • The air monitoring station using as reference does not measure SO2, CO2 and VOC. • Generally, CO2 is not measured in air monitoring station. • The concentration of SO2 is $\sim$ 0 ppb therefore we are not going to include in the measurements. • The data of VOC is limited to other air monitoring stations that don't measure CO, NO NOÂň2 and O3 and PM2.5. The VOC calibration would have required a separate study.

(3).

Fig1.

"The MAAQSbox holds five gas sensors and two particle sensors. The gas sensors included in the calibration are CO, CO2 O3, NO2, and NO (B4 sensors AlphaSense, Inc.)" _______________________________ (1). In both the Introduction (Line 80) and the Methods (Line 180) it is stated that the aim of this work is to 'evaluate' the performance of the sensors in their MAAQSbox, yet the design of their study and the data presented only reports a calibration (and not even the full calibration data), not an evaluation.

(2) • A new aim of this study was established including the reviewer comments for line 80 and 180.

(3)

Line 85. "We seek to relate sensor response, temperature, humidity, and concentrations other species exhibiting cross sensitivities to reference field measurements performs. Our analysis seeks to determine the accuracy and precision with which low-cost sensors perform during periodic "in-use" calibration."

Line 190. "The aim of the field calibration is to compare the low-cost sensor performance to a reference instrument in the field." _______________________________ (1). The co-location dataset comprises a single 6.5-day (154 hour) co-location at one time of year at one site. This dataset is then used to derive a multivariate linear regression calibration for each sensor against its relevant 'reference instrument' value using sensor signals and the internal airstream T and RH values as dependent variables. For calibration of the NO2 and O3 sensors, sensor signals from the other species sensor were also included to allow for potential cross-species interference.

(2)

• The field measurements were conducted during two different periods. The period 1 was during September-October 2018 and the period 2 was during March-April 2019.

Period 1 includes 154 hours of data for all sensors. Period 2 includes 244 hours for CO, 169 hours for NO, 86 for NO2, and 87 hours for O3. There are no PM2.5. data available for period 2. Calibration 1 is based on data collected during the first half, hours 1 to 76, of period 1. Calibration 2 is based on data collected during the second half of period 1, hours 77 to 154. Calibration 3 is based on the entire 154 hours of data collected during period 1. Calibration 4 is based on all the data available from period 2. • The stability of the calibrations was determined as follows: o Calibration 1, based on the first half of period 1 was tested against data from the second half of period 1. o Calibration 2, based on the second half of period 1 was tested against data from the first half of period 1. o Calibration 3, based on all 154 hours of period 1 was tested against data collected during the first half of period 1, and separately against data collected during the second half of period 1. o Calibration 4 is based on all the data available in period 2. The performance of calibration 3 and calibration 4 was tested against period 2 data. (3) Fig.2, Fig3, Fig4, and Fig5 _____ (1). However, the authors do not present the actual calibration equation coefficient values and their p values (they only state which variables are included in each sensor calibration equation).

(2) • A table with the p-values for Calibration 1, 2, 3, and 4 were added. Table S1, S2, S3, and S4. The Table S1 is shown as an example. (3)

See Fig.6 (Table S1).

Line 315 "Among the variables included in each model, the We signal presented the lowest p-values. Details of p-values are shown in Table S1, S2, and S3 in additional material."

Line 385 "Table S4 in additional material shows the p-values for Calibration 4. The We p-values for CO and NO, and O3 are lower than 0.01. The humidity p-values are lower than 0.01 for NO and O3. The temperature p-value is lower than 0.01 only for CO." _____ (1). Nor do they present visualisations and/or statistics for the raw comparisons of sensor values against respective reference concentrations. Consequently, in the ab-

sence of such information the reader is not able to gauge how well or not each sensor performs prior to the multivariate regression fits, i.e. to gauge how much modification to raw sensor output is being made by the derived multiple regression calibration equation. In other words the reader does not get a sense of how much the sensor signal needs to be corrected for the contribution of other variables to the signal, particularly the extent to which there has to be correction for cross-interference between the NO2 and O3 sensors. Such information would tell the reader how important other variables are.

(2). • Charts with the raw signal and explanations are added. • Table S1, S2, S3, and S4 in additional material show the p-value

(3) Line 260. "Before describing the performance of the various calibrations, it is useful to consider the stability of the primary raw signal of the sensors, We. The raw responses of the sensors during three time windows; the first half of period 1, the second half of period 1, and period 2 are plotted against AMS data in Figs.5 (a-d)." See Fig. 7 __________ (1). A more fundamental flaw, however, is that there is no independent evaluation of the calibration: the same data is used both to derive a calibration equation and then to justify the goodness of the calibration once applied to that data. If one derives a predictor equation from a dataset and then applies the predictor equation to exactly the same dataset then of course the predictions (and their 'evaluation' statistics) are likely to be very good. At the very least, there needs to be sufficient co-location data to (randomly) split into 'training' and 'test' sub-datasets in order to provide some (quasi)independent statistical evaluation of a derived calibration. More usefully still, what potential users of this MAAQSbox need to know is how well does a calibration equation hold in time and at different locations. Is there evidence of any long-term drift in sensor performance/ calibration?

(2) • This was answered above. ______

(1). If the sensors in the MAAQbox is calibrated at one location, does the same calibration hold at another location and/or at another time? If the MAAQbox is calibrated prior to a mobile deployment and is then used as intended on a mobile platform how well does its calibration hold up when the MAAQBox is co-located back at the reference monitoring station?

(2) • This was answered above.
* * *
(1). All that the data presented in this paper show is that an underlying relationship for sensor performance self-consistently holds within a single 154 hour period.

(2) • This was answered above.
* * *
Some additional comments: (1) The regression equation written on the panel of Figure 6c does not seem correct. The intercept appears to be much larger than 0.29 ppb, and eyeballing this panel suggests that the plotted regression line is giving much higher values for estimated NO2 than the stated regression line would predict; for example, for a reference value of 15 ppb the regression equation predicts a sensor value of 12.14 ppb but the plotted line shows higher estimated NO2 than this. Also, this panel should include the origin of the scatter plot.

(2) • It was an error in the calculation of the equation. The equation is y = 0.86x + 2.1. This is shown in Fig6 (c) in Calibration 3.

(3) See Fig.8
* * *
(1) The scatter plot in panel 6a should also include the origin of the plot, and why does the regression equation for this panel not have an intercept coefficient? Even if the coefficient is not statistically significant its value should be included to indicate that the regression included an intercept in the fit.
[Figure]

(2) • It was an error in the calculation of the equation. The equation is y = 0.96x + 0.02. This is shown in Fig.6 (a).

(3) See Fig9.

(1)The increased scatter in the calibration scatter plot for NO2 (Figure 6c) is noted but there is no discussion of this. Given that NO2 is a key pollutant in the urban environment, for which quantification by instruments such as MAAQSbox is most keenly sought, there needs to be further comment on what is underlying this poorer performance for NO2 measurement.

(2) • The P-value inputs for NO2 are presented in additional material. • The NO2 and O3 signals are presented in the paper. • Explanations for poor NO2 performance are presented in the paper.

(3) Line 340 "Poorer performances for NO2 has also been reported by other researchers where R2 = 0.4 to 0.89 (Castell et al., 2017; Mijiling et al., 2017). The poorer performance of the NO2 sensor compared to other low-cost sensors is because it is sensitive to environmental conditions, wind speed, impurities from the air, VOCS concentrations, sensor aging, and CO2 cross-sensitivity (Tian et al., 2019; Pang et al., 2018)."
* * *
(1). As indicated above, we are not given the magnitudes or p values of coefficients in the calibration equations: which one of the variables is having the most influence on the NO2 response during this deployment?

(2) • This was answered above.
* * *
[Figure]

**Fig. 1.**

**a)**

y = 0.957x + 0.0159
R² = 0.957

y = 0.9614x + 0.0191
R² = 0.9614

y = 0.9604x + 0.0095
R² = 0.9604

- Calibration 3
- Calibration 1
- Calibration 2

CO MAAQSBox [ppm] / CO AMS [ppm]

**b)**

y = 0.9488x + 0.6697
R² = 0.9488

y = 0.9727x + 0.5334
R² = 0.9727

y = 0.9881x + 0.3069
R² = 0.9881

- Calibration 3
- Calibration 1
- Calibration 2

NO MAAQSBox [ppb] / NO AMS [ppb]

**c)**

y = 0.9623x + 0.6716
R² = 0.9623

y = 0.801x + 2.2488
R² = 0.801

y = 0.8599x + 2.0447
R² = 0.8599

- Calibration 3
- Calibration 1
- Calibration 2

NO₂ MAAQSBox [ppb] / NO₂ AMS [ppb]

**d)**

y = 0.9855x + 0.162
R² = 0.9855

y = 0.9784x + 0.2669
R² = 0.9784

y = 0.9796x + 0.2769
R² = 0.9796

- Calibration 3
- Calibration 1
- Calibration 2

O₃ MAAQSBox [ppb] / O₃ AMS [ppb]

**e)**

y = 0.6158x + 3.2608
R² = 0.6158

y = 0.6019x + 2.0699
R² = 0.6019

y = 0.2153x + 1.4661
R² = 0.2153

- Calibration 3
- Calibration 1
- Calibration 2

PM₂.₅ MAAQSBox[μg m⁻³] / PM₂.₅ AMS [μg m⁻³]

**Fig. 2.**

a)

b)

c)

d)

e)

**Fig. 3.**

a) Plot: CO MAAQSBox [ppm] vs CO AMS [ppm]
- First Half Calibration 3
- Second Half Calibration 3
y = 1.0947x - 0.0046, R² = 0.8792
y = 0.9813x - 0.0084, R² = 0.9607

b) Plot: NO MAAQSBox [ppb] vs NO AMS [ppb]
y = 0.9768x + 1.3654, R² = 0.9833
y = 0.8758x + 0.8397, R² = 0.9284
- Second Half Calibration 3
- First Half Calibration 3

c) Plot: NO₂ MAAQSBox [ppb] vs NO₂ AMS [ppb]
y = 1.0272x + 0.3169, R² = 0.745
y = 0.8094x + 2.7842, R² = 0.9267
- Second Half Calibration 3
- First Half Calibration 3

d) Plot: O₃ MAAQSBox [ppb] vs O₃ AMS [ppb]
y = 0.934x + 0.9477, R² = 0.9782
y = 1.0308x - 0.3943, R² = 0.9828
- Second Half Calibration 3
- First Half Calibration 3

e) Plot: PM₂.₅ MAAQSBox [µg m⁻³] vs PM₂.₅ AMS [µg m⁻³]
- Second Half Calibration 3
- First Half Calibration 3
y = 0.4376x + 1.7443, R² = 0.172
y = 0.5817x + 2.8656, R² = 0.5836

**Fig. 4.**

[Figure]

Fig. 5.

Table S1

| Variable | CO | NO | NO$_2$ | O$_3$ | PM$_{2.5}$ |
|---|---|---|---|---|---|
| Humidity | 4.09E-02 | 6.43E-06 | 1.63E- | 9.44E-06 | 9.37E-02 |
| Temperature | 5.75E-02 | 6.70E-36 | 3.50E- | 5.97E-02 | 1.92E-02 |
| *We*_CO | 3.18E-49 | | | | |
| *Ae*_CO | 9.95E-11 | | | | |
| *We*_NO | | 2.51E-43 | | | |
| *Ae*_NO | | 3.80E-01 | | | |
| *We*_NO$_2$ | | | 2.99E- | 6.92E-35 | |
| *Ae*_NO$_2$ | | | 6.85E- | 4.05E-01 | |
| *We*_O$_3$ | | | 2.60E- | 1.48E-36 | |
| *Ae*_O$_3$ | | | 4.83E- | 2.01E-01 | |
| PM$_{2.5}$ | | | | | 3.02E-02 |

**Fig. 6.**

[Figure]

**Fig. 7.**

[Figure]

**Fig. 8.**

[Figure]

**Fig. 9.**

---

## Author Comment (AC3) · 15 Nov 2019

(1) The most significant flaw in the analysis presented is that it seems that the data used to train the calibration models are the same used to evaluate the same models? If so this is not a valid test, and the training and test data need to be independent data sets. (2)

 c The field measurements were conducted during two different periods. The period 1 was during September-October 2018 and the period 2 was during March-April 2019. Period 1 includes 154 hours of data for all sensors. Period 2 includes 244 hours for CO, 169 hours for NO, 86 for NO2, and 87 hours for O3. There are no PM2.5. data available for period 2. Calibration 1 is based on data collected during the first half, hours 1 to 76,

of period 1. Calibration 2 is based on data collected during the second half of period 1, hours 77 to 154. Calibration 3 is based on the entire 154 hours of data collected during period 1. Calibration 4 is based on all the data available from period 2. • The stability of the calibrations was determined as follows: o Calibration 1, based on the first half of period 1 was tested against data from the second half of period 1. o Calibration 2, based on the second half of period 1 was tested against data from the first half of period 1. o Calibration 3, based on all 154 hours of period 1 was tested against data collected during the first half of period 1, and separately against data collected during the second half of period 1. o Calibration 4 is based on all the data available in period 2. The performance of calibration 3 and calibration 4 was tested against period 2 data.

(3) See Fig2, Fig3, Fig4, and Fig5

(1) Overall the calibration approach is not clear, with no indication of the improvement achieved with the increasing complexity of the calibration equation used. It would be helpful to the reader if the authors could provide a baseline performance of the sensors using a simple linear fit to the raw sensor signals, before including other variables such as temperature. This would enable the impact of sensor interferences, e.g. from temperature, to be understood in both the laboratory and field calibrations. (2). • Charts with the raw signal and explanations are added (3) Line 265. "Before describing the performance of the various calibrations, it is useful to consider the stability of the primary raw signal of the sensors, We. The raw responses of the sensors during three time windows; the first half of period 1, the second half of period 1, and period 2 are plotted against AMS data in Figs.5 (a-d)"

See Fig7 ___

(1) As this is a description of a new instrument the authors should provide an assessment of the measurement uncertainty.

(2) • The uncertainty was measured by the root-mean-square error (RMSE) as shown in table 1.

(3) See Fig6. Line 315. "Also, adding the temperature and humidity terms only improves R2 and RMSE markedly in four cases, Calibration 1 NO, Calibration 3 NO, Calibration 2 NO2, and Calibration 3 NO2." Line 320 "Overall the sensor behaved poorly with RMSE values essentially equal to mean measured values." Line 345 "This relatively poor performance is consistent with the high RMSE and low R2 values reported in Table 1."

_____ (1) The poor performance seen for the OPC-N2 sensor when compared to reference measurements is not adequately discussed. Early studies using these sensors identified a significant humidity dependence impacting the data under high humidity conditions. A study by Antonio et al. (2018) developed a correction for this instrumental effect on the OPC-N2, resulting in an apparent improvement in data quality. The authors should at the very least acknowledge this earlier work and discuss the implications for the work presented here. (2) • We added humidity as part of independent variables. The p-values of the models for humidity, temperature, and OPC-N2 are in additional material. We also added references. (3)

Line 345. "Another research model achieves R2 =0.75 (Di Antoni et al., 2018) and ranges from 0.8 to 0.93 (Chatzidiakou et al., 2019). A measurement conducted in Memphis, TN presented diverse results range R2 from 0.52 to 0.81 (Feinberg et al., 2019)." ____ (1) Table 1 has no units on values other than the average mixing ratio.

(2) • We added the unit of each value in Table 1. (3) See Fig6

_____ (1) The statement on line 321 that calibrations will last _ 3 months has no supporting evidence and should either be removed or justified. References: Di Antonio A, Popoola OA, Ouyang B, Saffell J, Jones RL. Developing a relative humidity correction for low-cost sensors measuring ambient particulate matter. Sensors (Switzerland) 2018;18:2790. doi: 10.3390/s18092790. (2) • We added the reference.

(3) Line 450. "This supports a re-calibration periodically because of the sensitivity of sensor changes over time, which we anticipate to be ∼3 months (Di Antoni et al.,

2018).".

[Figure]

a)

CO MAAQSBox [ppm] vs CO AMS [ppm]

y = 0.957x + 0.0159
R² = 0.957

y = 0.9614x + 0.0191
R² = 0.9614

y = 0.9604x + 0.0095
R² = 0.9604

- Calibration 3
- Calibration 1
- Calibration 2

b)

NO MAAQSBox [ppb] vs NO AMS [ppb]

y = 0.9488x + 0.6697
R² = 0.9488

y = 0.9727x + 0.5334
R² = 0.9727

y = 0.9881x + 0.3069
R² = 0.9881

- Calibration 3
- Calibration 1
- Calibration 2

c)

NO₂ MAAQSBox [ppb] vs NO₂ AMS [ppb]

y = 0.9623x + 0.6716
R² = 0.9623

y = 0.801x + 2.2488
R² = 0.801

y = 0.8599x + 2.0447
R² = 0.8599

- Calibration 3
- Calibration 1
- Calibration 2

d)

O₃ MAAQSBox [ppb] vs O₃ AMS [ppb]

y = 0.9855x + 0.162
R² = 0.9855

y = 0.9784x + 0.2669
R² = 0.9784

y = 0.9796x + 0.2769
R² = 0.9796

- Calibration 3
- Calibration 1
- Calibration 2

e)

PM₂.₅ MAAQSBox[μg m⁻³] vs PM₂.₅ AMS [μg m⁻³]

y = 0.6158x + 3.2608
R² = 0.6158

y = 0.6019x + 2.0699
R² = 0.6019

y = 0.2153x + 1.4661
R² = 0.2153

- Calibration 3
- Calibration 1
- Calibration 2

**Fig. 1.**

a)

CO MAAQSBox [ppm] vs CO AMS [ppm]

- Second Half Calibration 1
- First Half Calibration 2

y = 0.9247x - 0.0684
R² = 0.9596

y = 1.0651x + 0.0264
R² = 0.8686

b)

NO MAAQSBox [ppb] vs NO AMS [ppb]

- Second Half Calibration 1
- First Half Calibration 2

y = 0.6997x + 9.0032
R² = 0.2352

y = 1.0353x + 4.2199
R² = 0.9796

c)

NO₂ MAAQSBox [ppb] vs NO₂ AMS [ppb]

y = 0.5903x + 1.9592
R² = 0.8479

y = 1.4069x - 2.6951
R² = 0.6394

- Second Half Calibration 1
- First Half Calibration 2

d)

O₃ MAAQSBox [ppb] vs O₃ AMS [ppb]

- Second Half Calibration 1
- First Half Calibration 2

y = 1.1254x - 0.8722
R² = 0.9805

y = 0.8069x + 2.9104
R² = 0.969

e)

PM₂.₅ MAAQSBox [μg m⁻³] vs PM₂.₅ AMS [μg m⁻³]

- Second Half Calibration 1
- First Half Calibration 2

y = -0.3908x + 14.811
R² = 0.0244

y = 0.5095x + 1.5782
R² = 0.5392

**Fig. 2.**

**a)**

Legend:
- First Half Calibration 3
- Second Half Calibration 3

y = 1.0947x - 0.0046
R² = 0.8792

y = 0.9813x - 0.0084
R² = 0.9607

CO MAAQSBox [ppm] vs CO AMS [ppm]

**b)**

y = 0.9768x + 1.3654
R² = 0.9833

y = 0.8758x + 0.8397
R² = 0.9284

- Second Half Calibration 3
- First Half Calibration 3

NO MAAQSBox [ppb] vs NO AMS [ppb]

**c)**

y = 1.0272x + 0.3169
R² = 0.745

y = 0.8094x + 2.7842
R² = 0.9267

- Second Half Calibration 3
- First Half Calibration 3

NO$_2$ MAAQSBox [ppb] vs NO$_2$ AMS [ppb]

**d)**

y = 0.934x + 0.9477
R² = 0.9782

y = 1.0308x - 0.3943
R² = 0.9828

- Second Half Calibration 3
- First Half Calibration 3

O$_3$ MAAQSBox [ppb] vs O$_3$ AMS [ppb]

**e)**

- Second Half Calibration 3
- First Half Calibration 3

y = 0.4376x + 1.7443
R² = 0.172

y = 0.5817x + 2.8656
R² = 0.5836

PM$_{2.5}$ MAAQSBox [µg m$^{-3}$] vs PM$_{2.5}$ AMS [µg m$^{-3}$]

**Fig. 3.**

[Figure]

**Fig. 4.**

[Figure]

a)
CO_We [Volts] vs CO AMS [ppm]
y = 0.3846x + 0.26, R² = 0.94
y = 0.3666x + 0.3272, R² = 0.87
y = 0.3148x + 0.266, R² = 0.93
CO_We_Period 2
CO_We_Period 1_First_Half
CO_We_Period 1_Second_Half

b)
NO_We [Volts] vs NO AMS [ppb]
y = 0.0004x + 0.2894, R² = 0.9298
y = 0.0004x + 0.3026, R² = 0.1305
y = 0.0004x + 0.2921, R² = 0.4939
NO_We_Period 2
NO_We_Period 1_First_Half
NO_We_Period 1_Second_Half

c)
NO₂_We [Volts] vs NO₂ AMS [ppb]
y = 0.0002x + 0.2348, R² = 0.5772
y = 0.0002x + 0.2346, R² = 0.7025
y = -8E-05x + 0.232, R² = 0.042
NO2_We_Period 2
NO2_We_Period 1_First_Half
NO2_We_Period 1_Second_Half

d)
O₃_We [Volts] vs O₃ AMS [ppb]
O3_We_Period 2
O3_We_Period 1_First_Half
O3_We_Second_Data
y = 7E-05x + 0.231, R² = 0.0603
y = 1E-04x + 0.2249, R² = 0.2953
y = -4E-05x + 0.2275, R² = 0.1225

e)
PM₂.₅ OPC-N2 [µg m⁻³] vs PM₂.₅ AMS [µg m⁻³]
PM2.5_Period 1_First_Half
PM2.5_Period 1_Second_Half
y = 0.09x + 1.6877, R² = 0.0631
y = 0.5183x - 0.0824, R² = 0.5501

**Fig. 5.**

**Table S1**

| Variable | CO | NO | NO$_2$ | O$_3$ | PM$_{2.5}$ |
|---|---|---|---|---|---|
| Humidity | 4.09E-02 | 6.43E-06 | 1.63E- | 9.44E-06 | 9.37E-02 |
| Temperature | 5.75E-02 | 6.70E-36 | 3.50E- | 5.97E-02 | 1.92E-02 |
| *We*_CO | 3.18E-49 | | | | |
| *Ae*_CO | 9.95E-11 | | | | |
| *We*_NO | | 2.51E-43 | | | |
| *Ae*_NO | | 3.80E-01 | | | |
| *We*_NO$_2$ | | | 2.99E- | 6.92E-35 | |
| *Ae*_NO$_2$ | | | 6.85E- | 4.05E-01 | |
| *We*_O$_3$ | | | 2.60E- | 1.48E-36 | |
| *Ae*_O$_3$ | | | 4.83E- | 2.01E-01 | |
| PM$_{2.5}$ | | | | | 3.02E-02 |

**Fig. 6.**